# Child-Owned Poultry Intervention Effects on Hemoglobin, Anemia, Concurrent Anemia and Stunting, and Morbidity Status of Young Children in Southern Ethiopia: A Cluster Randomized Controlled Community Trial

**DOI:** 10.3390/ijerph20075406

**Published:** 2023-04-05

**Authors:** Anteneh Omer, Dejene Hailu, Susan Joyce Whiting

**Affiliations:** 1School of Nutrition, Food Science and Technology, Hawassa University, Hawassa P.O. Box 5, Ethiopia; 2School of Public Health, Hawassa University, Hawassa P.O. Box 5, Ethiopia; 3College of Pharmacy and Nutrition, University of Saskatchewan, Saskatoon, SK S7N 5E5, Canada

**Keywords:** egg, poultry, hemoglobin, anemia, concurrent anemia and stunting, morbidity

## Abstract

Cereal-based diets contribute to anemia in Ethiopian children. Eggs have nutrients to boost hemoglobin levels as well as counter concurrent anemia and stunting (CAS) and morbidity status. A community trial, targeting 6–18 months old children, was conducted in Halaba. Two clusters were randomly selected and allocated to intervention (N = 122) and control (N = 121) arms. Intervention group (IG) children received egg-laying hens with caging in a cultural ceremony declaring child ownership of the chickens. Parents promised to feed eggs to the child. Health and agriculture extension workers promoted egg feeding, poultry husbandry, and sanitation to IG families. Control group (CG) had standard health and agriculture education. At baseline, groups were not different by hemoglobin, anemia, CAS, and morbidity status. Mean hemoglobin was 11.0 mg/dl and anemia prevalence was 41.6%. About 11.9% of children had CAS and 52.3% were sick. Using generalized estimating equations, the intervention increased hemoglobin by 0.53 g/dl (ß:0.53; *p* < 0.001; 95%CI: 0.28–0.79). IG children were 64% (*p* < 0.001; odds ratio [OR]:0.36; 95%CI: 0.24–0.54) and 57% (*p* = 0.007; OR: 0.43; 95%CI: 0.21–0.73) less likely to be anemic and have CAS, respectively, than CG, with no difference in morbidity. Child-owned poultry intervention is recommended in settings where anemia is high and animal-source food intake is low.

## 1. Introduction

Anemia remains a global public health challenge, affecting about 40% of children 6–59 months old [1]. Anemia has multifaceted causes, ranging from acute blood loss, nutrient deficiencies and infection, to autoimmune and genetic disorders. Nutrient deficiency and inflammation are the most common causes of anemia in children [2]. Among nutrients, particularly iron, vitamin B12, vitamin A, folate, zinc and protein deficiencies are associated with anemia [2,3,4]. Unfortunately, the diets of children under two of age in Africa and South Asia usually lack or poorly provide these nutrients [5,6]. Children in Africa, East Asia, and the Pacific and South Asia rank the highest in one or more of the deficiencies of iron, zinc, and vitamin A [7].

The most recent Ethiopian demographic and health survey (EDHS) reported that more than half of children below 5 years of age were anemic and, opposite to the global trend, the prevalence of anemia had increased compared to the preceding EDHS survey [8]. In Ethiopia, iron deficiency accounts for 25–36% of total anemia among pre-school children [9]. A recent study attributed iron deficiency and inflammation to cause 21% and 5% of total anemia among 6–59 months old children, respectively [10]. The cause of the remaining portion of anemia is not well-defined. Ethiopia set a target of decreasing anemia to 40% by the year 2025 [11].

The usual diets of children 6–24 months old in Ethiopia are cereal-based, rarely containing animal-source foods. Studies reported that, except for iron, dietary intakes of zinc, folate, vitamin B12, vitamin A, and protein among Ethiopian children are generally inadequate [9,10,12]. Due to the poor protein and micronutrient content of complementary foods, high demand for nutrients and frequent episodes of diseases together with environmental and other factors, under two years of age children are the most vulnerable group to anemia and its far-reaching consequences [2]. Anemia greatly impairs the immune system, thus increasing the risk of morbidity and mortality among young children. It also impairs neuro-cognitive and motor development hampering school performance and productivity [2].

Nutrition interventions have had encouraging results in alleviating anemia among infants and young children. Home fortification of children’s diet using micronutrient sprinkles [13,14,15] and lipid-based nutrient supplements [16,17,18,19] are well-documented strategies of anemia reduction, improvement of iron status, and hemoglobin levels. Nutrition education that improved infant feeding practices significantly decreased anemia and increased hemoglobin compared with the existing intervention [20] and WASH/Malaria intervention [21].

Reports from food-based strategies are inconsistent. Caterpillar cereal compared with the usual diet in Democratic Republic of Congo increased hemoglobin significantly [22]. Locally produced food products containing germinated amaranth, maize, small fish, and edible termites in Kenya decreased hemoglobin and iron status [23] compared with corn-soy blend (CSB), most likely due to the iron bioavailability inhibition effect of the amaranth. Rice-based complementary foods with small fish and edible spiders in Cambodia compared with CSB showed no effect on iron status [24]. A trial that compared meat with multi-micronutrient-fortified rice-soy cereal reported no difference in anemia rates but significantly lowered the iron deficiency among the cereal group [25]. After providing one egg daily for six months in Malawi, no difference was found with the control group (no egg) in the prevalence of anemia, iron deficiency and iron deficiency anemia, hemoglobin levels, and iron status [26]. Little research has been done in Ethiopia on the effects of nutrition interventions on anemia among infants and young children.

We recently implemented a nutrition-sensitive child-owned poultry intervention that provided chickens and caging materials, declaring ownership to the children with the aim of increasing egg intake and minimizing disease risk from direct contact with the birds and their feces. Poultry husbandry was improved, and egg intake was increased in the intervention group (mean weekly egg intake 4.85 vs. 0.4; *p* < 0.001) [27]. In addition, at six months, underweight and stunting reduced; weight for age and weight for height z-scores increased and motor skills (running, kicking ball, and throwing ball milestones) were attained at an earlier age significantly compared to the control group [28].

Our pilot study conducted in 2016 recorded a marginally significant reduction of anemia (RR = 0.48; 95%CI = 0.24–0.96) after providing a gift of egg-laying hens to children and promoting separate chicken sheltering [29]. Although awareness of disease risk related to chicken production was enhanced and the practice of keeping chickens separately improved significantly compared to the baseline and control group, only a quarter of intervention households were able to have and utilize cages. Due to economic reasons, most cages were made from farm leftover stalks of maize and sorghum that were not strong enough to last long [30]. Observational studies in Sub-Sahara Africa associated livestock ownership including poultry with anemia and lower hemoglobin [31,32] although this was not always the case [33]. Taking the challenges related to the construction of separate chicken shelters observed in the pilot study and the probable negative effect of rearing chickens on child health and nutrition outcomes including masking of intervention effects into consideration, this project provided cages beside chickens to minimize the disease risk from direct contact with the birds and their feces. This paper reports on the intervention effects on hemoglobin, anemia, and morbidity status of young children.

## 2. Materials and Methods

### 2.1. Study Design, Participants, and Sample Size

With a 6-months follow-up period, a cluster randomized and controlled community trial (Trial registration: NCT 03355222) was conducted from May to November 2018 in Halaba district, Southern Nations Nationalities and Peoples Region (SNNPR), 85 km southwest of Hawassa, the regional capital. Halaba is classified as midland with an average elevation of 1800 m above sea level. Livelihood mainly depends on rainfed farming mixed with livestock production. Maize, sorghum, teff, haricot bean, wheat, millet, and chili are the major agricultural products of the district. Due to erratic and irregular rainfall, Halaba is frequently affected by drought and is endemic to malaria. [34]. The district is divided into 10 catchments (clusters), each having 1 health center that oversees health posts that function in kebeles/villages inside the catchment. Health Extension Workers (HEWs) based in health posts provide health and nutrition services to the community with the support of Health Development Team Leaders (HDTLs) who are volunteers networking at a ratio of 1 for every 30 households.

From the 10 clusters in the district, 2 were randomly selected and allocated to intervention and control wings. For matching reasons, a kebele/village was purposely chosen from each selected cluster. Distance from their respective catchment health center and existing health, nutrition, and agriculture interventions were taken into consideration during village selection. The kebeles selected for the study do not have a common marketplace. In addition, they are located in opposite directions (east and west) to the district administrative center, Halaba Kulito town.

The study was powered to observe the effect of the intervention on hemoglobin levels. We did not find trials that assessed the effect of egg intake on hemoglobin among under two years old children except for one, which evaluated the consumption of four egg yolks per week (excluding the egg white) and recorded no significant effect on hemoglobin [35]. Hence, a medium effect size (0.5) was entered in the calculation with a design effect of 2, power of 80%, and 10% loss to follow-up, which provided a sample size of 126 mother-child dyads for each study group. After excluding those who were sick and taking drugs, on nutritional treatment or severely and acutely malnourished, and reported by their caregivers to be allergic to eggs, a total of 253 apparently healthy 6–18 months old children who lived for at least 6 months in the selected kebeles were enrolled in the study (Intervention Group, IG, [N] = 127, Control Group, CG, [N] = 126). Detailed information on sampling, participants’ enrolment, and study protocol has been published [27].

Participant families, HEWs, HDTLs, and agriculture extension workers (AEWs) in both study villages were not informed about the existence of the two study groups. Data collectors deployed for baseline and end line surveys were different and not informed regarding the allocation of study groups. Medical laboratory technologists who collected samples as well as carried out lab analysis were not aware of any study groups. Samples sent to the laboratory for analysis were uniquely coded with no identifiable information about where they came from and the name of the subject.

### 2.2. Intervention

The intervention had three core components: local capacity building, chicken and caging gift, and social and behavior change communications (SBCC), and these have been previously described in detail [27]. In brief, HEWs, AEWs, and HDTLs, who were frontline implementers at the kebele level, received two days of training on the benefits of egg consumption for children’s health and nutrition, the risk of disease transmission from free-range chickens and its related possible negative effects, the importance of cage utilization and environmental sanitation and improved poultry husbandry, and their role in the study. Workers in the control kebele received the usual nutrition and agriculture training provided to them with their position.

Following the capacity-building training, religious leaders, community elders, and village management team in the intervention kebele were sensitized about the objective and approach of the research project and their role in the study, paving the way to the implementation of Chicken and Caging Gift Ceremony (CCGC). The CCGC is an innovative community-based culture and religion-sensitive approach that empowers children to be owners of chickens and the eggs that they produce, aiming for one-egg-a-day consumption. AEWs organized the gift ceremony in collaboration with HEWs and HDTLs in the kebele farmer’s training center (FTC). Religious leaders declared ownership of the chickens and all the eggs they would produce to the children and marked the practice of selling or sharing the eggs with anyone or the chickens as “Haram”, an Arabic term meaning forbidden. Families promised to add at least two more hens, replace them if the birds die, not to sell or share the eggs and to feed the chicken-owner child one egg a day. Parents signed and received two egg-laying local breed hens, vaccinated for Newcastle disease, on behalf of the children. Two types of cages were also provided. One was a night coop that is locally made, lightweight to move around, easily cleanable, and can accommodate up to eight chickens. The other cage was an enclosure (25 square meter) where the chickens spend the day roaming for food and rest. Wood logs, mesh wire, and nails were provided, and families built the cage in their own compounds after learning how to by creating a model with AEWs at the kebele FTC. The AEWs also provided orientation on utilizing and cleaning the night coop.

After the gifting of chickens and caging was done, HEWs held a demonstration of two types of egg preparations at the health post: hard-boiled and smashed egg yolk for children 6–7 months old and boiled and smashed whole egg for children above 7 months. Following HEWs’ instructions, caregivers boiled and smashed the egg (egg yolk or whole egg depending on their child’s age), expressed breast milk to soften the preparation for easy swallowing and fed their children during the session.

Having dietary, anthropometry and child-owned poultry production assessment findings, the HEWs and AEWs provided tailored individual counselling every month using SBCC cards prepared for this purpose based on their area of specialty. In their counselling, they promoted one egg-a-day consumption, child-owned poultry production, cage utilization, improved poultry husbandry, and environmental sanitation. HDTLs also passed the SBCC messages to the households under their network. Any contact with caregivers including monthly data collection, home visits, and caregivers’ visits to the health post for health service was used as an opportunity to promote baby-friendly chicken production and egg feeding.

### 2.3. Ethics Approval

The study was approved by Hawassa University, Ethiopia, and the University of Saskatchewan, Canada. Formal communication was made with district and kebele-level health, agriculture, and administration offices. Consent was obtained from all caregivers to participate in the study. They were also informed of lab tests and procedures before sample collection. They received a bar of soap and oil as compensation for their time during the follow-up. Children excluded from the study due to reports of egg allergy were equally treated by being rewarded with chickens like the participant children. Children in the control group also received egg-laying hens along with CCGC messages but only after the end line survey.

### 2.4. Measurements

Baseline information was collected on sociodemographic and economic characteristics, infants and young child feeding practices, livestock production, and poultry husbandry practices. The intervention effect on hemoglobin and anemia status was the primary outcome of this study, while morbidity status was measured as a secondary outcome. Effects on child-owned poultry production, egg intake, anthropometry, and gross motor skills development were reported previously [27,28].

Two medical laboratory technologists measured hemoglobin concentration using a portable spectrophotometer (Hemocue Hb 301; HemoCue AB, Ängelholm, Sweden) and the presence of antigens of *P. falciparum* and *P. vivax* using rapid diagnostic test CareStart™ Malaria HRP2/pLDH (Pf/Pv) Combo, Kit/25 (Access Bio Inc., Addis Ababa, Ethiopia) at baseline and end line. Tests were carried out in the field from finger-prick blood samples following the manufacturer’s instructions.

Formol-ether concentration technique was run to test intestinal helminthiasis from randomly selected sub-samples (50 children) from each group following standard operating procedures presented by Cheesbrough [36]. Stool samples were transported to Halaba General Hospital Laboratory Parasitology Unit for analysis after being emulsified in 10% formol water. At the laboratory, the formol-fecal suspension was strained first, 10% formol water was added, and it was centrifuged for 1 min. To the sediment, 8 mL of formol water and 4 mL of diethyl ether were added and centrifuged. Then, the sediment was examined under a microscope for intestinal helminths with egg count.

Morbidity symptoms of fever, coughing, vomiting, and diarrhea as well as skin, eye, and ear infections that occurred in the last two weeks before the date of assessment were recorded at baseline and monthly during follow-up based on caregivers’ reports. Diarrhea was defined as the passage of three or more loose or liquid stools per day (or more frequent passage than is normal for the individual) [37]. Caregivers were informed to report vomiting when their child forcefully threw up food; not spit out small amounts of food. Fever was defined as hot to the touch. A child was recorded to have a cough if the cough, regardless of the duration, came with one of these symptoms: a runny nose or cold, trouble breathing, fever (hot to touch), wheezing sound, and being constant (several times in a day) or stayed for more than two days with no additional symptoms [38]. The presence of one of these symptoms was reported as a skin infection: itching rash with small papules, visible lesions, fluid-filled vesicles and blisters, abscess, ringworm (itchy circular lesion with a fine scaly area on body and scalp), and scabies [39]. Indications of ear infection included the presence of pus draining from one or both of the ears or the child being irritable and rubbing his/her ear [40]. Eye infection/disease was defined as having one of these symptoms: discharge from the eyes; eyelids that are stuck together after waking from sleep; red eyes/eyelids and swollen eyelids; visible white or gray sore on the iris; white foamy lesions on the conjunctiva [41,42].

### 2.5. Data Analysis

Hemoglobin values were adjusted for altitude before analysis as per WHO guideline [43]. Baseline characteristics were presented with descriptive analysis and comparisons between groups were made by chi-square and independent samples *t*-test for categorical and continuous variables, respectively. Considering the cluster randomization design of the study and some missed subjects during follow-up, the Generalized Estimating Equations (GEE) model was primarily utilized to evaluate intervention effects on hemoglobin levels and status of anemia, concurrent anemia and stunting (CAS), and morbidity. GEE of linear and binary logit models was run for continuous and categorical variables, respectively. Stratified analysis was also conducted to evaluate intervention effects among specific groups. Effect sizes were reported as beta (β), odds ratio (OR), and relative risk (RR). *p*-values < 0.05 were taken as statistically significant. Data were processed using IBM SPSS version 28 (Chicago, IL, USA).

## 3. Results

A total of 243 children completed the study and entered into the analysis. It was necessary to exclude seven children who were lost to follow-up and three children who were found to be sensitive to eggs. A flow diagram has been previously published [27].

### 3.1. Baseline Characteristics

Except for child sex (X^2^ = 8.77; *p* = 0.003), socio-economic and demographic characteristics including poultry production and husbandry, infant and young child feeding practices and maternal characteristics were not significantly different among the study groups (Table 1). Detailed information has been previously published [28]. Groups were also comparable at enrollment by their nutritional, malaria, intestinal helminthiasis, total anemia, and morbidity status.

### 3.2. Intestinal Helminthiasis and Malaria Infection

No child was positive for intestinal helminth at baseline and malaria at baseline and end line in both groups. About 7% of the children had helminth infection at end line with no group difference (*p* = 0.436). Except for one child who was diagnosed with *Ascaris lumbricoides* (220 eggs/gram of feces), the others were infected with *Hymenolepis nana* (egg count ranging from 2 to 19 eggs/gram of feces). No multiple helminth infections were found.

### 3.3. Hemoglobin, Anemia, and Concurrent Anemia and Stunting (CAS)

Among the total children, mean hemoglobin concentration at baseline was 11.0 g/dl. Anemia prevalence was 41.6% of which mild, moderate, and severe anemia constituted 61.4%, 36.6%, and 2%, respectively. About 11.9% of the children were both anemic and stunted (CAS). Mild anemia was more prevalent in the intervention group (X^2^ = 4.063; *p* = 0.044) as moderate anemia was in the control (X^2^ = 2.654; *p* = 0.103). GEE showed that hemoglobin was significantly increased by 0.53 among intervention children compared to the control at the end line (Table 2) after adjusting for baseline weight for age z-scores. Child age and sex did not modify the effect on hemoglobin. The mean hemoglobin level among children in the control group declined at six months, which was accompanied by significantly increased anemia and CAS prevalence. On the contrary, children in the intervention group were 64% and 57% less likely to be anemic and concurrently affected by anemia and stunting, respectively, compared to children in the control.

Stratified analysis showed that mean hemoglobin levels increased among children in the intervention group at six months regardless of baseline nutritional status. Significant increases in mean hemoglobin concentrations were observed only among children with normal nutritional status (normal weight, not stunted, and not wasted) compared to their matches in the control (Table 3). Underweight, stunted, and wasted children at baseline were too few to evaluate the effect of the intervention on hemoglobin. Effect sizes found in all strata of normal nutritional status were almost the same (β ranging from 0.53–0.54 and 95% CI = 0.24–0.28; 0.81–0.82). Hemoglobin concentrations improved in both intervention and control groups at six months among children anemic at baseline, but the incremental increase was significantly higher in the intervention arm (β = 0.83; 95% CI = 0.47, 1.19) than in the control. In contrast, mean hemoglobin concentrations declined in both study groups among children who were non-anemic at baseline; however, hemoglobin decreased less in the intervention group than in the control. From all strata analyzed, the highest hemoglobin increase was observed among anemic children.

Regardless of the baseline nutrition, anemia, and CAS status of the children, the prevalence of anemia and CAS decreased at endline in the intervention group while it increased in the control (Table 3 and Table 4). More than 4 out of 10 and nearly 1 out of 5 children in the control group who had no anemia and CAS at enrollment developed anemia and CAS, respectively, by six months. The odds of developing anemia and CAS among non-anemic and non-CAS children at baseline was reduced by 75% and 78%, respectively, in the intervention group compared to control. About 70% of anemic children and more than 80% of children with CAS at enrollment in the intervention group no longer had anemia and CAS, respectively, at 6 months. On the contrary, more than 80% and 69% of children affected by anemia and CAS, respectively, at baseline in the control group remained classified as anemic and having CAS at end line.

### 3.4. Morbidity Symptoms

More than half of the children in both groups exhibited one or more morbidity symptoms at baseline that remained unchanged at six months. Groups were not different in morbidity symptoms (fever, cough, diarrhea, and vomiting) and infections of the skin, eye, or ear at baseline and end line. Adjusted analysis (child sex and age, baseline anemia and nutritional status, and baseline hemoglobin levels) also did not find a significant difference between groups at end line. However, reductions in episodes of vomiting and eye, ear and other infections were recorded at end line in both groups (Table 5). Among the total children, older ages were found to become sick less (showing any morbidity symptom or skin, eye, ear, and other infection) at end line (*p* = 0.026; β = −0.040; 95% CI = −0.076, −0.005).

## 4. Discussion

Our innovative nutrition-sensitive poultry intervention that enabled children to be owners of chickens improved hemoglobin concentrations and reduced the prevalence of anemia and concurrent anemia and stunting at six months. The intervention has resulted in significantly improved poultry husbandry practices, particularly cage utilization, increased egg intake and improved nutritional status and attainment of developmental milestones, which has been published previously [27,28].

The mean hemoglobin of the children was higher and anemia prevalence was lower at baseline than that of children under two years old who participated in the 2016 EDHS: Hgb = 10 ± 1.63 g/dl; anemia prevalence of 71.9% [44]. The baseline anemia prevalence was comparable with what was found among 6–23 months old children who lived in rural Ethiopia in the midland agroecology [45]. Our intervention increased hemoglobin by 0.53 g/dl, unlike the egg trial in Malawi, which provided eggs to 6–9 months old children for daily consumption (Mazira project) and reported no improvement in hemoglobin concentration, ferritin, soluble transferrin receptor, and body iron index. No difference was also seen among egg and no egg groups in anemia, iron deficiency, and iron deficiency anemia prevalence [26]. The high burden of iron deficiency at enrolment and sustained inflammation during the study period was presented as the reason why egg intake could not improve the iron and anemia status of the children. In addition, the inhibition effect of phosvitin and ovotransferrin found in the egg yolk on nonheme iron from other foods and the high fish intake in both groups might have also hampered the intervention effect [46].

It is important to note the context difference between the Mazira project and our project. In the Mazira children, iron deficiency is found concurrently with 93% of the anemia and more than 98% to 100% prevalence of iron intake inadequacy was recorded at midline and end line [26,47]. Although we did not measure iron status, children in our project had higher hemoglobin levels and lower anemia prevalence than the Mazira children at baseline; most likely indicating better iron status. Another important difference to note is that children in the Mazira project had adequate baseline protein intake from complementary foods, including fish and breast milk [47]. About 32.8%, 8.7%, and 5.5% of children in our study had cow milk, egg, and legumes and nut intake, respectively, and no child had an intake of flesh foods on the day before the baseline survey. Moreover, the inhibition effect of egg yolk proteins might be minimum or nil in our study as caregivers were trained to feed the egg on its own, although they were free to prepare the egg in boiled or fried form as long as it was cooked well. Feeding eggs mixed with other foods was not reported at all during monthly counselling as well as home visits. In our pilot study, mixing eggs with other foods (mostly boiled and smashed potato) was reported by caregivers as a remedial action when they felt their child was sensitive to eggs [29]. Three children were found to be sensitive to eggs in the current study and we excluded them from the analysis of intervention effect on health and nutrition status.

Recent evidence revealed that iron deficiency accounted for 21% of anemia while folate and infection accounted for 6% and 5%, respectively, among the under-five population of Ethiopia [10]. Gashu et al. (2016) reported a very low prevalence of iron deficiency (9.1%) and iron deficiency anemia (5.3%) among children consuming a predominantly unrefined plant-based diet [48]. Cereals in Ethiopia have high iron content [49] and provide adequate iron, even under the assumption of low bioavailability (5%) [50]. This is partially due to extrinsic iron from soil contamination during harvesting and threshing of the grains [51,52] combined with the fermentation process during food preparation, which increases the bioavailability [53,54].

However, protein and micronutrient intakes, particularly vitamin A, zinc, folate, and vitamin B12, are very low [10,12]. Nearly two-thirds of children under five years old in Ethiopia’s southern region have inadequate protein intake, which is the lowest compared to the other regions [12]. The Ethiopia national food consumption survey also reported that the inadequacy of protein intake was associated with low consumption of flesh foods and legumes, and having an intake of foods that were poor in protein content and amino acid composition. This is in agreement with our dietary intake findings. Based on the existing evidence, micronutrients (other than iron), and protein deficiencies are important causes of nutritional anemia among Ethiopian children.

Protein intake has been associated with improved hemoglobin status. Valine, leucine, and isoleucine (branched chain amino acids (BCAA)) were reported to be positively and significantly correlated with levels of hemoglobin and anemic individuals were found to have significantly lower serum BCAA concentrations than non-anemic ones [55]. Protein deficiency is associated with reduced iron incorporation into hemoglobin [56], and low consumption of protein-source foods was associated with iron deficiency anemia in Ethiopia [57]. Improved amino acid intake increases hematopoiesis [58]. Egg white protein, particularly ovalbumin, has significant benefits in recovery from iron deficiency anemia in animal models [59]. Egg consumption substantially contributes to dietary adequacy of protein, fat, pantothenic acid [47,60,61,62,63], energy [64], phosphorus, vitamin D [61,62,63], vitamin B12 [47,61], biotin, vitamin E, cholesterol [61], choline, and its metabolites including docosahexaenoic acid (DHA) [62,63,65], riboflavin, selenium, vitamin A [47,62], α-linolenic acid, lutein + zeaxanthin, and potassium [62]. Provided that iron intake is adequate in our context, despite the source, we deemed that the increased egg intake improved the protein and micronutrient status of the children, which ultimately increased hemoglobin concentrations and reduced anemia and CAS prevalence [27]. We previously showed that the effect of whole egg intake was also observed in improving the nutritional status and motor development of the children [28].

The magnitude of CAS in this study population at baseline (11.9%) was lower than the national CAS rate among 6–23 months (23.9%) [66]. However, CAS rate in the control group reached the national level at end line (24%). Comparable prevalence (25.3%) was recorded among 6–23 months children in two districts from north and south Ethiopia [45] and pooled proportion of CAS among 6–59 months from 43 low and middle-income countries (21.5%) [67]. The study area and age difference might be the reason for the lower CAS prevalence at enrollment in our study population. CAS prevalence decreased significantly in the intervention group at six months. This might be due to the increased protein and micronutrient intake from the eggs as well as the consumption of vitamin A-rich fruit and vegetable intake, which significantly increased in the intervention group as a result of sales of excess eggs [27]. Analysis of the 2016 DHS data revealed that increased protein intake and consumption of vitamin A-rich fruit and vegetable intake were associated with lower odds of CAS in Ethiopia [66].

Although anemia and stunting are two different body conditions, they have communal determinants and pathways that are important to consider when designing interventions. Undernutrition causes both anemia and stunting as infection exacerbates both conditions [2,68]. Age and sex [66,69,70], low dietary diversity [70,71,72], not achieving minimum meal frequency [45], multiple micronutrient deficiencies [73], rural residence, and low household wealth (living in poorer households) [66,67,69], infection [66,69], drinking unsafe water [70,71], and household food insecurity [70,72,74] are important determinants of both stunting and anemia. Anemia and stunting are strongly associated with each other [75]. Stunted children have a higher risk of anemia than non-stunted ones and vice versa [76]. Stunting is a predictor of anemia [45,72,77] and vice versa [71,78]. Thus, interventions that improved the determinants or the causes communal to stunting and anemia might be effective in reducing CAS, as observed in our nutrition-sensitive child-owned poultry project that improved dietary diversity, macro-and micronutrient intake, and nutritional status.

Livestock ownership, such as sheep and goats, was significantly and positively associated with child anemia in Ghana, while free-range poultry showed marginal significance [32]. In Ghana, using population data, the household ownership of chickens, but not other animal species, was associated with higher odds of anemia among under-five years old children [31]. A systematic review reported that poultry production interventions, in contrast to observational studies, had modest benefits on anemia [79]. Chicken ownership was not associated with lower hemoglobin concentrations among children in Sub-Saharan Africa [80]. A recent study in Ghana found lower odds of anemia among children from households owning cattle, small livestock (goats, sheep, or pigs), and poultry than those owning no livestock [33]. We reported a marginally significant reduction of anemia in our pilot project [29] in which less than 6% of intervention households used separate chicken shelters at baseline, which increased to 25% at 6 months [30]. In the current study, we found a very significant reduction of anemia and CAS and increased hemoglobin concentrations with much bigger effect sizes. Besides increased egg intake, the improved poultry husbandry practice particularly cage utilization might have contributed to this result. Almost all households in the intervention group utilized cages and kept chickens separately day and night throughout the study period [27], which might have minimized the contact of the children with the birds and their feces, although we did not measure this variable. Campylobacter infection transmitted from poultry is associated with environmental enteric dysfunction, a chronic sub-clinical condition that causes loss of nutrients by inhibiting absorption and leaking out nutrients [81]. While a study in Ethiopia found no harmful effects of the provision of chickens on child anemia [82], it is recommended to use cages to keep chickens away from children and minimize fecal contamination of the home environment.

The child-owned poultry intervention showed no significant effect on reducing morbidity symptoms. Our pilot study [29] and the Ecuador egg trial [83] also did not find an effect on the reduction of morbidity symptoms. Vomiting was seen more frequently in the pilot study treatment group, which might be due to egg sensitivity. About 7.5% of the children (*n* = 18) in the intervention group had developed egg sensitivity. In the current study, only three children were found to be sensitive to egg. The Ecuador trial [83] recorded diarrhea reports more in the egg group, which might have been related to egg intake, which is not the case in this study as diarrhea almost equally increased in both groups at end line. The absence of increased prevalence of morbidity symptoms and the improvements observed in the reduction of symptoms such as fever and cough (although not significant) indicate that there was no increased risk of disease in the intervention group that received chickens and produced more chickens [27]. This might be due to the practice of cage utilization practiced by all households combined with enhanced awareness of the disease risk related to chicken production and increased nutrient intake of the children through eggs. A study in Ethiopia found no evidence of any harmful effects of the provision of chickens on child morbidity, therefore supporting our findings [82].

To our knowledge, this is the first study to report the effect of egg intake on reducing concurrent anemia and stunting. Much emphasis was placed on sustainability in the design and implementation of the intervention, which is the strength of this study. The approach of the intervention is unique and has the potential to sustain the poultry production and egg-feeding behavior observed during the study period. However, the study had limitations. The sample size did not allow us to conduct sub-group analysis, particularly among children who were wasted and underweight at baseline, as they were too few in number to conduct comparisons between the groups. If we had measured the children’s exposure to chicken feces, fecal contamination of the household environment, and environmental contamination, the study would be able to evaluate the role of cage utilization in reducing morbidity as well as improving anemia status. As morbidity symptoms were based on caregivers’ reports, there would be a recall bias. In addition, understanding the operational definitions of morbidity symptoms among caregivers might be different despite the explanation given, which may result in over-reporting or under-reporting of symptoms. The six months study duration might not be enough to observe the intervention effect on reducing morbidity symptoms.

## 5. Conclusions

A nutrition-sensitive child-owned poultry intervention significantly increased the hemoglobin concentrations and reduced anemia and concurrent anemia and stunting prevalence among under two years old children over six months of implementation. This has a huge advantage, particularly for low-income countries that suffer from both conditions well-known for their far-reaching negative consequences. In an Ethiopian context, where iron intake is less concerning but protein and several micronutrient inadequacies are high, whole egg consumption can play a key role in filling the gap as they provide almost all the essential nutrients that support early life. Combined with its potential for sustainability and model suitability for rural communities in low-income countries with low animal-source food intake by children, the high effect sizes recorded within six months make this approach a plausible strategy to combat anemia among infants and young children who are the most vulnerable groups. Large-scale implementation of the model or integration with existing interventions is warranted in rural settings where animal-source food intake is low.

## Figures and Tables

**Table 1 ijerph-20-05406-t001:** Selected baseline characteristics and IYCF practices.

Description		Intervention (N = 127)		Control (N = 126)	
		N	%	N	%
**Livestock production**					
Poultry production	Chicken	26	20.5	33	26.2
Chicken care	Day cage/separated place ^a^	3	11.5	2	6.1
	Night shelter/cage ^a^	5	19.2	5	15.2
**Maternal Characteristics**					
Age in years	Mean age (SD)	27.3 (4.68)		27.5 (4.18)	
Education on feeding eggs	Received	51	40.2	52	41.3
Awareness of chicken feces as risk to child	Aware	30	23.6	40	31.7
**Child Characteristics**					
Sex	Female *	46	36.2	69	54.8
Age (month)	Mean (SD)	10.9 (3.18)		11.4 (4.28)	
**IYCF**					
Breastfeeding	Currently fed on breastmilk	125	98.4	122	96.8
Complementary food (CF)	Currently on CF	120	94.5	117	92.9
	Mean age of introduction: months (SD)	6.13 (0.59)		6.2 (0.69)	
Egg intake history	Ever fed	65	51.2	57	45.2
% children fed with egg	24h before survey	10	7.9	12	9.5
	The week before survey	30	23.6	35	27.8
Eggs consumed per week	Mean (SD)	0.23 (0.42)		0.29 (0.51)	

* Statistically significant (*p* < 0.05). IYCF = infant and young child feeding. ^a^ Computed among chicken owners (denominator is 26 for intervention and 33 for control group).

**Table 2 ijerph-20-05406-t002:** Intervention effect on hemoglobin, anemia, and concurrent anemia and stunting (CAS).

	Baseline		End Line		Unadjusted		Adjusted	
	IG (N = 122)	CG (N = 121)	IG (N = 122)	CG (N = 121)	Effect Size ^1^		Effect Size ^1^	
	Mean (SD)	Mean (SD)	Mean (SD)	Mean (SD)	β (95% CI) ^a^	*p*	β (95% CI) ^a^	*p*
Hgb	11.1 (1.1)	10.9 (1.4)	11.4 (1.0)	10.6 (1.4)	0.52 (0.26, 0.77)	<0.001	0.53 (0.28, 0.79) ^†^	<0.001
	N (%)	N (%)	N (%)	N (%)	OR (95% CI) ^b^	*p*	OR (95% CI) ^b^	*p*
Anemia	50 (41)	51 (42.1)	26 (21.3)	70 (57.9)	0.45 (0.30, 0.68)	<0.001	0.36 (0.24, 0.54) ^††^	<0.001
CAS	16 (13.1)	13 (10.7)	8 (6.6)	29 (24)	0.52 (0.29, 0.94)	0.031	0.43 (0.23, 0.80) ^†††^	0.007

IG = Intervention group; CG = Control group; Hgb = Hemoglobin; CAS = Concurrent anemia and stunting; OR = Odds ratio. Independent, auto-regressive, and exchangeable working correlation matrices provided the same results. ^1^ Computed by Generalized Estimating Equations (GEE) linear (^a^) and binary logit (^b^) models. ^†^ Adjusted for baseline weight for age z-scores. ^††^ Adjusted for baseline hemoglobin values. ^†††^ Adjusted for baseline length for age, weight for age, and weight for length z-scores.

**Table 3 ijerph-20-05406-t003:** Intervention effect on hemoglobin and anemia among groups stratified by baseline nutrition, anemia and CAS status.

Strata	N		Hemoglobin				Anemia			
			Baseline	End Line	Effect Size ^1^		Baseline	End Line	Effect Size ^2^	
			Mean (SD)		β (95% CI)	*p*	N (%)		OR (95% CI)	*p*
Normal Weight	IG	100	11.2 (1.1)	11.4 (1.0)	0.54 (0.26, 0.81)	<0.001	39 (39.0)	21 (21.0)	0.40 (0.26, 0.64)	<0.001
	CG	97	10.9 (1.3)	10.6 (1.3)			41 (42.3)	59 (60.8)		
Not Stunted	IG	87	11.2 (1.1)	11.4 (1.1)	0.53 (0.24, 0.82)	<0.001	34 (39.1)	19 (21.8)	0.38 (0.23, 0.62)	<0.001
	CG	84	11.0 (1.1)	10.6 (1.4)			38 (45.2)	52 (61.9)		
Not Wasted	IG	111	11.2 (1.1)	11.4 (1.0)	0.54 (0.28, 0.81)	<0.001	44 (39.6)	23 (20.7)	0.43 (0.28, 0.65)	<0.001
	CG	111	10.9 (1.4)	10.6 (1.4)			48 (43.2)	64 (57.7)		
Anemic	IG	50	10.1 (0.8)	11.3 (1.0)	0.83 (0.47, 1.19)	<0.001	50 (100.0)	15 (30.0)	0.11 (0.04, 0.26) ^a^	<0.001
	CG	51	9.7 (1.3)	10.1 (1.2)			51 (100.0)	41 (80.4)		
Non-Anemic	IG	72	11.8 (0.6)	11.5 (0.9)	0.27 (0.03, 0.51)	0.026	0 (0)	11 (15.3)	0.25 (0.12, 0.57) ^a^	0.001
	CG	70	11.7 (0.6)	11.0 (1.0)			0 (0)	29 (41.4)		
Non CAS	IG	106	11.3 (1.0)	11.4 (1.0)	0.44 (0.20, 0.69)	<0.001	34 (32.1)	21 (19.8)	0.41 (0.26, 0.65)	<0.001
	CG	108	11.4 (1.0)	10.7 (1.4)			38 (35.2)	61 (56.5)		

IG = Intervention group; CG = Control group; CAS = Concurrent anemia and stunting; OR = Odds ratio. ^1^ Computed by GEE linear model. ^2^ Computed by GEE binary logit model except those denoted by (^a^) that the odds ratio was calculated by binary logistic regression. ^1,2^ Working correlation matrices of independent, auto-regressive, and exchangeable provided similar results.

**Table 4 ijerph-20-05406-t004:** Intervention effect on CAS among the groups stratified by baseline nutrition, anemia, and CAS status.

Strata	N		Baseline	End Line	OR (95% CI) ^1^	*p*
			N (%)			
Normal weight	IG	100	11 (11)	6 (6)	0.28 (0.11, 0.74)	0.010
	CG	97	5 (5.2)	18 (18.6)		
Not wasted	IG	111	15 (13.5)	7 (6.3)	0.22 (0.09, 0.53)	0.001
	CG	111	12 (10.8)	26 (23.4)		
Stunted	IG	35	16 (45.7)	4 (11.4)	0.17 (0.05, 0.58)	0.005
	CG	37	13 (35.1)	16 (43.2)		
Not stunted	IG	87	0 (0)	4 (4.6)	0.26 (0.08, 0.84)	0.025
	CG	84	0 (0)	13 (15.5)		
Anemic	IG	50	16 (32.0)	5 (10.0)	0.20 (0.07, 0.61)	0.004
	CG	51	13 (25.5)	18 (35.3)		
Non-anemic	IG	72	0 (0)	3 (4.2)	0.23 (0.06, 0.88)	0.031
	CG	70	0 (0)	11 (15.7)		
CAS	IG	16	16 (100)	3 (18.8)	0.10 (0.02, 0.57)	0.010
	CG	13	13 (100)	9 (69.2)		
Non-CAS	IG	106	0 (0)	5 (4.7)	0.22 (0.08, 0.61)	0.003
	CG	108	0 (0)	20 (18.5)		

IG = Intervention group; CG = Control group; CAS = Concurrent anemia and stunting; OR = Odds ratio. ^1^ Computed by binary logistic regression.

**Table 5 ijerph-20-05406-t005:** Effect of nutrition-sensitive child-owned poultry intervention on the prevalence of morbidity symptoms.

Morbidity	Baseline		End Line		OR (95% CI) ^1^	*p*
	IG (N = 122)	CG (N = 121)	IG (N = 122)	CG (N = 121)		
	%	%	%	%		
Any symptom	51.6	52.9	54.1	58.7	0.81 (0.62–1.04)	0.102
Fever	35.2	28.1	27.0	31.4	0.81 (0.61–1.08)	0.144
Cough	27.0	19.8	29.5	27.3	0.95 (0.70–1.28)	0.736
Diarrhea	18.0	14.9	20.5	17.4	0.79 (0.56–1.10)	0.158
Vomiting	16.4	14.9	6.6	9.9	0.77 (0.52–1.17)	0.220
Skin Infection	7.4	4.1	7.4	6.6	1.10 (0.66–1.86)	0.709
Eye, ear and other infections	9.8	12.4	1.6	4.1	1.27 (0.72–2.23)	0.413

^1^ Computed by GEE binary logit model with exchangeable correlation matrix. Independent and auto-regressive correlations provided very close results to exchangeable matrix. No significant difference was found among groups in all working correlations matrices.

## Data Availability

Data will be made available upon request.

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
