# Peer review of "Child-Owned Poultry Intervention Effects on Hemoglobin, Anemia, Concurrent Anemia and Stunting, and Morbidity Status of Young Children in Southern Ethiopia: A Cluster Randomized Controlled Community Trial"

_ijerph, 2023, doi:10.3390/ijerph20075406_

Round 1

Reviewer 1 Report

Thanks for this interesting paper, describing the impact of egg consumption in young children.

Nevertheless I have some doubts.

What is known about the consumption of these young children? Looking at babies and young children, I doubt about the real intake of the eggs. It is high.

Further, I would suggest to reduce the conclusions to a more simple result, no politics or sentences about thing you have not investigated.

Detail comments:

L 81. Is this the amount of eggs offered or consumed? I can hardly believe that such young children will or can eat almost 1 egg per day.

L 300. A dot is missing: 0.54

L 326. Choose the same wording for the description of Table 3 and 4. In T.4 the word Intervention is not mentioned.

L 337-338. I would skip this. Donot mention non-significant results.

L 349. I donot understand, how cgildren with an age of less than 1 year can be “owners of chickens”. What do you really mean with this? How can babies be “owners”?

L 362. Please be more precise here, what is meant by “complementary feeding behaviors” as well as “complementary foods”. It is vague now.

L 364: What is a nutrition BCC intervention? What has been done? What is fed differently?

L 368. Again, what do you mean with these words? “complementary and responsive feeding”? What has been done? What is fed differently?

L 355-L376. I would move everything, which is not about egg-intervention to the introduction. Please discuss only the findings on egg-interventions. And please discuss differences and/or similarities in the outcomes, which helps to understand the positive outcomes of your own study. Like you do in L 377 forward.

L377. Children of 6-9 month and daily egg consumption?? Those children hardly eat anything besides breast milk. Than it is a big word “daily egg consumption”. So, maybe there are no differences between the two groups, because these very young children hardly can eat any additional food, egg included.

L 384. Again “the high fish intake”? Is this serious, because babies donot eat much fish. So, what is high?

L 419. “foods poor in protein content”?? Is this a low protein content? Or a poor or inadequate protein composition?

L423. Hemglobin >> hemoglobin.

L 431-440. Shouldn’t you be more precise on the impact differences of egg yolk versus egg white? What did your children consume? Some young children do not like the boiled egg yolk. The yolk has most of the fats and fat soluble vitamins.

L43-439. If you would generalize these findings, is the result due to a better protein intake, protein with a better amino acid profile? Could it either be from meat, organ meat or insect protein as well? This seems to me important for the discussion. So, general lack of protein or is it protein specific?

L445. Forthy-three >> 43

L448-450. This could be a huge confounding factor. How will you deal with this statistically? Why did not use you correlation statistics? Pearson correlations?

L468 positively or negatively associated?

L474-476. Why is this not immediately after L468-69? This is also about Ghana and livestock type.

L483. Why do you think this is? What can be causal, if you keep chickens separated? Is this separation really the cause for a better child growth?

L492-495. None of these were significant, which means there is no difference. If so, donot discuss this.

L496-97. Was vomiting significant? If not, skip this as well.

L505. “produced more chickens” ??? As meat animals? As eggs, what does this mean?

L511-512. Vague, what do you mean exactly with sustainability in designd, and what is the cause and effect explanation?

L514. What is “egg feeding behavior” Do you mean it gives the parents the possibility to feed more eggs to their children?

L516. I think being underweight or wasted is the same? Please use only one of them.

L520-523. This type of discussion does not make sense. Skip this. There is much more, what you have not measured.

L531. Your conclusions are far too long and donot cover what your title says. It should mainly contain sentences about what eggs are doing in relation to health of this group of children. Very specific. Donot speculate about “large scale implementations” etc, because that is not what you investigated, compared or whatever. You have shown that the uptake of laying hens improved the food security, improved most probably the protein supply and reduced anemia and stunting. The rest is speculation. You cannot say something about “the best solutions”, there is no comparison of different poultry systems or comparisons made with milk, goats, etc

Further please explain words and sentences, because it says nothing or is in crowd language (see below).

L532. “nutrition-sensitive child-owned” This is abracadabra, and should be reformulated in actions.

L533. What is the difference between “anemia” and “concurrent anemia”? Suggestion: skip the last.

L534-535. “its potential for sustainability” plus “model suitability” ?? These are words with multiple explanations and should be described was has been done.

L541-542. Again: lots of words, which are hard to understand. Please describe what you have done and what actions should/could be made.

L542. You donot know, if these are “the best alternatives”, because you did not compare different applications.

L543-544 This is the main message/conclusion to my opinion

Author Response

Response to reviewer 1 comments is attached

Reviewer 2 Report

This is an interesting and well-written manuscript that aimed to evaluate the impact of an innovative nutrition-sensitive poultry intervention that allowed children to be owners of chickens on hemoglobin, anemia and morbidity status of young children in Ethiopia. Considering the high prevalence of anemia in sub-Saharan Africa, including Ethiopia, and the short and long-term consequences of anemia, programs that have potential in reducing anemia rates and improving nutrition overall are important and should be encouraged. This manuscript is clear and well-written and the findings are interesting and make an important contribution to the field. A couple of comments and suggestions for improving the manuscript are as follows:

1.     Recruitment: How many participants (children 6-18mo) were allowed per household or per family? Was there a criteria to include only one child per household? Or you could recruit as many children 6-18 mo within a particular household?

2.     Line 149-150 states: “workers in the control group received the usual nutrition and agricultural training provided to them with their position”. Who provided this training? Was it the research team? Or is this something that is already in existence (standard of care) in the communities. In other words, would a worker have received this training had they not been a part of your study community?

3.     So for the control communities, apart from the capacity building training (the “usual” nutrition and agriculture training) provided to the workers, households did not receive any intervention during the study period correct?

4.     Lines 158-160: “Sinful” sounds a bit extreme! I’d suggest you re-word this to use another term – maybe “not following protocol”? or “wrong doing”? or “mis-guided”.

5.     Lines 159-160: Sharing with who? Are you referring to inter-household or intra-household sharing? Please be more specific.

6.     Lines 158-160: I am curious about something. Is this practice (the religious ceremony) common in Ethiopia? Would it be something that they’d culturally not frown upon? Or is it a usual cultural or religious practice?

7.      Lines 158-160: Was the aim of the religious ceremony to increase adherence to the study protocol and also consumption of eggs by the children?

8.     165-166: How big was the enclosure such that chickens could roam and search for food?

9.     Lines 176-178: Were the messages provided during counselling based on each individual’s situation? Or were the messages general (standard messages) to everyone? This is not too clear.

10.  Lines 181-184: Was there a standard frequency of messaging? Is it possible that some households received more messages than other households?

Author Response

Response to Reviewer 2 comments is attached
